# COVID-19 Pandemic Has Changed the Psychiatric Profile of Adolescents Attempting Suicide: A Cross-Sectional Comparison

**DOI:** 10.3390/ijerph20042952

**Published:** 2023-02-08

**Authors:** Rebeca Gracia-Liso, Maria J. Portella, Joaquim Puntí-Vidal, Elena Pujals-Altés, Jordi Torralbas-Ortega, Marta Llorens, Montserrat Pamias, Marc Fradera-Jiménez, Itziar Montalvo-Aguirrezabala, Diego J. Palao

**Affiliations:** 1Department of Mental Health, Parc Taulí-University Hospital of Sabadell, 08208 Barcelona, Spain; 2Department of Psychiatry and Forensic Medicine, Unitat de Neurociència Traslacional I3PT-Inc., Autonomous University of Barcelona (UAB), 08193 Barcelona, Spain; 3Sant Pau Mental Health Research Group, Sant Pau Biomedical Research Institute (IIB Sant Pau), 08041 Barcelona, Spain; 4Department of Psychiatry, Hospital de la Santa Creu i Sant Pau, 08041 Barcelona, Spain; 5Department of Psychiatry and Forensic Medicine, Autonomous University of Barcelona (UAB), 08193 Barcelona, Spain; 6Centro de Investigación Biomédica en Red de Salud Mental (CIBERSAM), 28029 Madrid, Spain; 7Nursing Care Research Group, IIB Sant Pau, 08041 Barcelona, Spain; 8Child and Adolescent Psychiatry and Psychology Department, Hospital Sant Joan de Déu of Barcelona, 08950 Barcelona, Spain; 9Institut d’Investigació i Innovació Parc Taulí I3PT-CERCA, 08208 Sabadell, Spain

**Keywords:** adolescent suicide, COVID-19, mental health, suicide attempt

## Abstract

The COVID-19 pandemic is having a major impact on the mental health of adolescents, leading to suicidal behaviors. However, it remains to be clarified whether the COVID-19 pandemic has changed the psychiatric profile of adolescent suicide attempters. Methods: a retrospective observational analytical study was conducted to assess age, gender and clinical characteristics of adolescents attempting suicide during the year before and the year after the global lockdown. Results: ninety adolescents (12–17 y.o.) were recruited consecutively from February 2019 to March 2021 at the emergency ward for having attempted suicide. Fifty-two (57.8%) attended before the lockdown (pre-pandemic group) and thirty-eight (42.2%) the year after (pandemic group). There were significant differences in diagnostic categories between the periods (*p* = 0.003). Adjustment and conduct disorders were more frequent in the pre-pandemic group, while anxiety and depressive disorders were more prevalent during the pandemic. Although the severity of suicide attempts did not show significant differences between the two study periods (0.7), the generalized linear model showed that the suicide attempt severity was significantly associated with current diagnosis (*p* = 0.01). Conclusions: the psychiatric profile of adolescents attempting suicide was different before and during the COVID-19 pandemic. During the pandemic, the proportion of adolescents with a prior psychiatric history was lower, and most of them were diagnosed with depressive and anxiety disorders. These diagnoses were also associated with a greater severity in the intentionality of suicide attempt, regardless of the study period.

## 1. Introduction

Suicide is a public health issue and one of the leading causes of mortality among the adolescent population [1], being the third cause of death in people aged 10–19 in Spain [2]. The characteristics of suicide in adolescents differ from adults, as psychological and social risk factors seem to play an important role in addition to biological factors [3]. Risk factors that have been associated with suicidal behavior in adolescents include: the presence of mental disorders [4,5,6], family history of suicide [7], personality traits, particularly impulsivity [8], and trauma and adverse life events [9,10]. Acute stressors act as precipitating factors for suicidal behavior among adolescents who are vulnerable due to their previous circumstances. The presence of mental health disorders is one of the main risk factors of suicide in adolescents. Several studies have linked depression with an increased risk of ideation and suicide attempt in adolescents [11,12], as well as other disorders such as anxiety disorders, personality disorders, and post-traumatic stress disorder [13]. Externalizing disorders, such as attention deficit hyperactivity disorder and conduct disorders, have been suggested to be more specific suicidal risk factors for children and adolescents [14,15], while the presence of psychiatric comorbidities would confer a greater risk of suicidal behavior [16,17]. As in adults, prior suicide attempts are the main risk factor for suicide and the best predictor of suicidal behavior [18,19,20].

The Catalan Health Department launched a specific suicide risk program (CRS programme from Catalan “Codi Risc Suïcidi”) in 2014 [21], which is a secondary prevention programme aiming to ensure the specialised follow-up of individuals at a high risk of suicide. The CRS has been running ever since, and adolescents who attempt suicide are referred to the programme by emergency services of hospitals in Catalonia, where they are visited within 72 h by a specialised practitioner in intensive interventions to reduce the risk of further suicide attempts.

The World Health Organisation declared the outbreak a global pandemic in March 2020. Lockdowns were initially adopted as a measure to contain the virus in affected countries. In Spain, a state of emergency was declared on 14 March 2020, confining all inhabitants to their homes, suspending all non-essential work activities, and closing schools and other educational facilities. The stay-at-home lockdown was maintained until the epidemiological situation improved. Less restrictive measures were adopted by the end of June 2020, allowing non-essential work activity to resume, while schools reopened in September 2020. However, due to the persistence of the pandemic, several social distancing and protection measures remained until late 2021 to prevent the spread of the virus.

There is evidence that the COVID-19 pandemic is having a negative psychological and social impact on the general population [22,23]. A number of studies show that the pandemic is associated with increased stress, anxiety, depression, and risk of suicide [24]. A recent meta-analysis found that suicide ideation, suicide attempts, and self-harm have increased during the COVID-19 pandemic (relative to event rates from pre-pandemic studies), with younger people and women appearing most vulnerable to suicide ideation [25]. The mental health consequences of the pandemic are expected to persist for a long time to come, and suicide prevention efforts will need to be heightened during and after the pandemic [26].

Evidence also suggests that the COVID-19 pandemic is having a negative impact on adolescent mental health, particularly in terms of depression and anxiety, due to the increased stress caused by the pandemic itself and the consequences of pandemic control measures [27]. So far, there is little evidence of the effects of the pandemic on suicidal behavior among children and adolescents. An early study in Japan concluded that suicide rates remained unchanged during the first wave [28]. However, more recent studies [29] warn of an increase in suicide rates in Japan among young people under 20 during the second wave of the pandemic, coinciding with the reopening of schools after the initial lockdown period. Another study conducted in Texas (USA) reported an increase in suicide attempts and suicidal ideation among young people aged 11–21 during the first 6 months of the pandemic compared to the same period the previous year [30]. A study in Philadelphia reported an increase in depression and suicide risk among adolescent girls during the pandemic year compared to the previous year [31]. In Spain, data from the CRS programme suggest that there was an increase in the number of suicide attempts by young people aged 12–19 during the months of the pandemic compared to the previous year, with a 195% increase in the number of suicide attempts by adolescent girls in the second phase of the pandemic, when schools were reopening [32].

Considering the increase in anxiety and depression disorders in adolescent populations associated with the COVID-19 pandemic, and since depression and anxiety disorders constitute one of the main risk factors for suicide, it remains to be clarified whether the COVID-19 pandemic might have changed the psychiatric profile of adolescent suicide attempters. We hypothesize that suicide attempts in adolescents will be more associated with diagnoses of depression and anxiety during the COVID-19 pandemic than pre-pandemic, and the previous history of psychiatric disorders of adolescents will also be different between these two periods. Moreover, these differences may account for the severity of the intention of suicide attempts.

## 2. Materials and Methods

### 2.1. Participants and Procedure

A retrospective observational cross-sectional analytical study was conducted to assess adolescents (aged 12–18) seen for a suicide attempt at the Psychiatric Emergency Department at Parc Tauli Hospital who agreed to participate. No replacements were accepted. Parc Tauli Hospital is the referral facility for psychiatric hospitalisation for the entire population of the Vallés Occidental Region in Catalonia, which includes 180,000 minors. The study was initially designed to investigate related factors of suicidal behavior in adolescents. With the appearance of the COVID-19 pandemic, this secondary study was designed, which took advantage of the routine clinical assessment performed in the facility so as to compare some clinical characteristics of participants before and during the pandemic. Inclusion criteria were: (1) adolescents aged 12–17 (inclusive) referred to the CRS programme following a suicide attempt. Exclusion criteria: (1) having a cognitive or neuropsychological impairment preventing assessment; (2) refusing to participate in the study; and (3) language barriers.

Sample recruitment and data collection took place from February 2019 to March 2021. Data collection was carried out by a clinician (a psychiatrist or psychologist) from the CRS programme working at the Child and Adolescent Mental Health Department.

### 2.2. Ethical Considerations

The original study was approved by the hospital’s ethics committee, and the current secondary study was conducted after an add-on to the original protocol which was also approved. All procedures were performed in accordance with the ethical principles set out in the Declaration of Helsinki (last version, Fortaleza 2013) [33]. All participants and their legal guardians (either parents or tutors) gave their informed consent in writing after receiving information about the study and its objectives, as well as assurances of confidentiality and personal data protection. Participation in the study did not involve any financial compensation.

### 2.3. Instruments

To explore changes in the study variables among adolescent patients who attempted suicide in the 12 months prior to the COVID-19 pandemic and those who did it during the first 12 months of the pandemic, we defined two time periods:-The pre-pandemic period, which includes participants recruited from the start of the study until the state of emergency was declared in Spain (February 2019–February 2020).-The pandemic period, which includes participants recruited from the declaration of the state of emergency in Spain to the end of the study (March 2020–March 2021).

Through a semi-structured interview different variables were gathered. For the purpose of this secondary analysis, age and sex, together with other clinical variables were used, such as: a psychiatric diagnosis according to DSM-5 criteria at the time of assessment; psychiatric comorbidities according to DSM-5 criteria at the time of assessment; personal psychiatric history including history of suicide attempts and self-harm, family history of suicidal behavior, and history of treatment with psychotropic drugs. The seriousness of the current suicide attempt was measured through the Beck’s Suicidal Intent Scale (SIS). The SIS is a 15-item questionnaire to be administered by a professional and designed to assess the severity of the suicide attempt associated with an episode of deliberate self-harm [34]. Each item is scored between 0 and 2, and the total score ranges from 0 to 30. There are not validated cut-off points. The higher the score, the more serious. The SIS has been validated in Spanish and the internal consistency of the total SIS was equal to 0.8 [35].

### 2.4. Data Analyses

Statistical analyses were performed using the IBM SPSS software (IBM Corp. Released 2017. IBM SPSS Statistics for Windows, Version 25.0. Armonk, NY, USA: IBM Corp.). Normal distribution was checked for quantitative variables by means of skewness and kurtosis. Those quantitative variables not normally distributed were to be analyzed with non-parametric tests. Mean and standard deviation were displayed. Qualitative variables were analyzed with chi-square or Fisher’s exact test; the number of participants and their percentage (*n*, %) were used for each category. To test the hypotheses of the study, comparisons between pre-pandemic and pandemic groups for all the relevant variables were carried out. A generalized linear model was run as it allowed us to build interaction terms. The model would include those variables that may show differences between the two study periods to determine their impact on the severity of the suicide attempt. The level of significance was set at 0.05.

## 3. Results

A total of 90 participants were included in the present study. Quantitative variables were normally distributed (age: skewness = −0.397 and kurtosis = −0.850; SIS: skewness = 0.676 and kurtosis = −0.092). Therefore, analyses including these variables were performed using parametric tests (ANOVA).

As can be seen in Table 1, there were no significant differences between individuals of the two study periods (pre-pandemic and pandemic groups) except for age, in which adolescents of the pandemic group were significantly younger than the pre-pandemic ones, though this is not practically relevant.

In relation to the primary psychiatric diagnosis of participants at the time of the study, statistically significant differences were identified between the pre-pandemic group and the pandemic group (see Table 1), i.e., a significantly higher percentage of unspecified adjustment disorders and conduct disorders were observed in the pre-pandemic group, while depressive and anxiety disorders were more prevalent in the pandemic group. Statistically significant differences were also identified in terms of psychiatric comorbidities between the two subgroups, with a greater presence of comorbid conduct disorder in the pre-pandemic group than in the pandemic group. Regarding prior psychiatric history, up to 60% of individuals had a psychiatric condition before the suicide attempt in the pre-pandemic period compared to less than 40% during the pandemic period (χ^2^ = 4.6; *p* = 0.03). In detail, the pre-pandemic group had significantly higher percentages of prior history of diagnosis of conduct disorder. By contrast, a significantly higher percentage of history of diagnosed anxiety disorder was detected in the pandemic group. See Table 1 for a summary of means, percentages, and statistics.

The presence of prior suicide attempts was more frequent in the pre-pandemic group, up to double the amount, although it was not statistically significant, while prior non-suicidal self-injuries were equally distributed between the two periods. There were no differences in the severity of the suicide attempt measured using the SIS between the individuals of the two periods.

Before running the generalized linear model, current diagnosis was recategorized into three categories because two levels of the variable only had one individual, and this could affect the overall distribution of data, leading to undesirable artefacts. Based on the two-dimensional internalizing/externalizing problems common in the conceptualization of mental health during adolescence [36], the new categories grouped anxiety and depressive disorders (with more internalizing problems); autism spectrum disorder and conduct disorder (with more externalizing problems); and maladaptive personality and adjustment disorder (as personality disorders). The generalized linear model included current diagnosis, comorbid conduct disorder, history of anxiety and conduct disorders, age, study period (pre-pandemic and pandemic), and period × current diagnosis. The model was statistically significant (Likelihood Ratio χ^2^ = 18.55; df = 9; *p* = 0.03; AIC = 567.53) and the severity of the suicide attempt was explained using a current diagnosis (Wald χ^2^ = 10.99, *p* = 0.004), while neither the rest of the factors nor the interaction of period × current diagnosis (see Table 2 for details) were statistically significant.

Indeed, there were significant differences in the SIS scores among the three diagnostic categories (F = 7.29; df = 2, 87; *p* = 0.001). Least Square Difference posthoc analyses showed significant differences in all pairwise comparisons except between personality and externalizing disorders (*p* = 0.051). See Figure 1 for the mean values of the SIS scores in the two groups, for all diagnostic categories and for grouped categories, see (3).

## 4. Discussion

Our research confirms that adolescents who attempted suicide during the pandemic period present more diagnoses of depression and anxiety than adolescents who attempted suicide in the year before the pandemic. In the pre-pandemic group, unspecified adjustment disorders, maladaptive personality traits, and conduct disorders were predominant. During the pandemic, a higher prevalence of major depressive disorder and anxiety disorders was observed, while the prevalence of maladaptive personality traits remained similar in both periods. This change in diagnoses could be linked to the increase in anxious and depressive symptoms in the adolescent population described in other studies on the pandemic [37,38]. The severity of the suicide attempt in our sample was explained using the current diagnosis and not by the study period, i.e., adolescents with depressive and anxiety disorders have a greater severity of suicidal intent than adolescents with other diagnoses, regardless of the period (pre-pandemic vs pandemic lag-time). Therefore, the pandemic did not directly imply an increase in the seriousness of the intentionality of the attempts, although during the pandemic there was a higher prevalence of anxiety and depressive disorders [39]. Particularly for anxiety disorders, a recent meta-analysis with 29 studies worldwide showed that the pooled prevalence in children and adolescents during the COVID-19 pandemic was estimated as 20.5%. Ref. [40] represents a significant increase compared to a 2015 study that reported a 6.5% prevalence in children and adolescents [41]. It has been reported that anxiety disorders increase the risk of subsequent suicidal thinking by eight times, and the risk of suicidal behavior by six times [42]. As for major depression, it has been estimated that one in four youths globally were experiencing clinically elevated depression symptoms during the COVID-19 pandemic [39], in accordance with our results. Mental suffering, especially depression, may have doubled during the COVID-19 pandemic [40], being the most common suicide causality in adolescents as reported by Manzar and colleagues [43]. The increase in anxiety and depression disorders during the pandemic, and the risk posed for suicidal behavior in adolescence, would be behind the raise of diagnoses in anxiety and depressive disorders in adolescents who attempted suicide during the pandemic in our sample. However, our findings did not show an increase in the number of attended adolescents at the emergency wards during the pandemic period; in fact, it found the opposite. This decrease is in line with the study carried out in Catalonia, in which a 6.3% decrease in suicide attempts was observed in 2020 compared to the previous year [44], related to the effects of lockdown during the first months and less consultation with health services. However, a more recent study in the same population and in which a longer period is studied (until June 2022) reports an increase in suicidal behavior during the COVID-19 pandemic, especially in women and in the youngest [45].

It should be noted that there has been a decrease in the diagnoses of conduct disorders in patients who attempted suicide during the pandemic, being more prevalent in the pre-pandemic group, both for current diagnosis at the time of assessment and for previous history and psychiatric comorbidity. Several studies have reported a decrease in the diagnosis of externalizing disorders during the COVID-19 pandemic [46,47]. Some authors suggest that the stability of externalizing disorders during the pandemic could be explained by the reduction in social and academic demands during this period [48].

It is important to highlight that in the pandemic period, there were fewer adolescents with a previous psychiatric history than in the pre-pandemic. This implies that, in the wake of the pandemic, attempting suicide has been one of the warning signs of emerging mental health problems among presumably resilient youths. In relation to this, the population study by Lehmann in Norway indicates that internalizing problems were more marked among the presumably least vulnerable youths, and that in those with higher levels of mental problems, the situation seemed more stable [46], according to our results.

Multiple studies have highlighted the negative impact of social isolation on adolescent mental health, resulting from government restrictions to prevent the spread of the virus [49], with the duration of isolation being a predictor of future mental health problems, especially depression [50]. This is particularly relevant in the context of the COVID-19 pandemic, as adolescents have been forced to maintain social distancing for long periods of time. Loneliness, because of the lockdown, is particularly harmful to adolescents, who are highly dependent on peer relationships for social development and emotional support [51]. Evidence shows that children and adolescents are more likely to experience high rates of depression and anxiety during and after months or years of being subjected to isolation and predicts self-harming, suicide ideation, and more lethal suicide attempts [52]. Social isolation, stay-at-home lockdowns, and feelings of uncertainty have been associated with increased anxiety, which is attributable to the disruption of extracurricular and social activities [53].

### Study Limitations

The main weakness of this study is the limited sample size, since the numbers correspond to a unique hospital in a middle-size town. This could undermine the statistical credibility of the results and the generalization of the findings. In any case, it represents one example of the impact of COVID-19 upon suicide attempts. Finally, this study was not designed to ascertain whether there is a causal relationship between the observed trends and the COVID-19 pandemic, as longitudinal studies with larger samples are needed to establish causal relationships.

## 5. Conclusions

The results point towards a different psychiatric profile of adolescents attempting suicide before and during the COVID-19 pandemic. During the pandemic, less than 40% of adolescents who attempted suicide had a prior psychiatric history, while for the rest the current attempt was a possible manifestation of the newly diagnosed disorder (mostly depressive and anxiety disorders), and the gateway to mental health services. In addition, adolescents with a diagnosis of depression and anxiety at the time of the suicide attempt had a greater severity in the intentionality of the attempt, with no differences between study periods.

Our results indicate that the pandemic has influenced the mental health of adolescents, causing mental disorders in previously healthy people, which without this exceptional situation, may not have occurred. This is especially serious when, as in our sample, a suicide attempt is the first detectable manifestation of the underlying disorder, thus delaying intervention and clouding the prognosis.

It is essential that new ways are found to help adolescents adjust and cope with changes in the context of the pandemic. For instance, the following measures could promote adolescent mental health during a pandemic: reducing the impact of forced physical distancing by maintaining the structure, quality, and quantity of social networks; promoting their sense of belonging to a group; counselling families so that they can provide emotional support to adolescents during lockdown periods using effective communication and supervising the maintenance of healthy routines. On the other hand, it is a priority to work on improving access to mental health services and on the detection of depressive and anxiety disorders in future pandemics, in order to carry out earlier treatments and thus reduce the risk of suicide. These actions could be key in preventing mental disorders in the adolescent population and reducing their risk of suicidal behavior.

## Figures and Tables

**Figure 1 ijerph-20-02952-f001:**
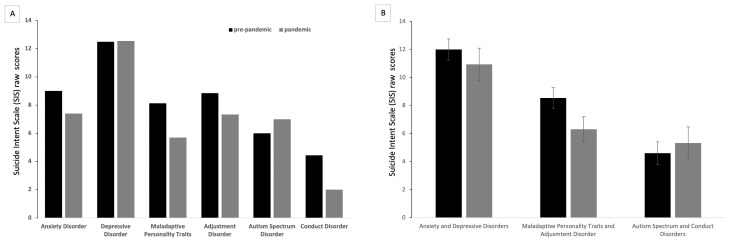
(**A**) Bar chart with the mean scores for Suicide Ideation Scale (SIS) in the two groups (pre-pandemic and pandemic periods) via diagnosis. (**B**) Bar chart with the mean scores and standard error of the mean (SEM) for SIS via diagnoses as grouped for the generalized linear model.

**Table 1 ijerph-20-02952-t001:** Demographics and clinical variables of the participating adolescents with a suicide attempt, divided via the two study periods (pre-pandemic from February 2019 to February 2020 and pandemic form March 2020 to March 2021). SD = Standard deviation; ADHD = Attention Deficit and Hyperactivity Disorder; SIS = Suicidal Intent Scale.

Variables	Total (*n* = 90) (100%)	Pre-Pandemic (*n* = 52) (58.8%)	Pandemic (*n* = 38) (42.2%)	*p*-Value
Age (mean ± SD)	15.2 ± 1.4	15.5 ±1.2	14.7 ±1.6	**0.009**
Sex (%fem)	76 (84.8%)	44 (84.6%)	32 (84.2%)	0.958
**Primary diagnosis at the time of assessment**				
Anxiety disorder	6 (6.7%)	1 (1.9%)	5 (13.2%)	**0.003**
Depressive disorder	17 (18.9%)	6 (11.5%)	11 (28.9%)	
Maladaptive personality traits	25 (27.8%)	15 (28.8%)	10 (26.3%)	
Unspecified adjustment disorders	26 (28.9%)	20 (38.5%)	6 (15.8%)	
Autism spectrum disorder	5 (5.6%)	1 (1.9%)	4 (10.5%)	
Conduct disorder	11 (12.2%)	9 (17.3%)	2 (5.3%)	
**Psychiatric comorbidities at the time of assessment**				
Depressive disorder	8 (8.9%)	5 (9.6%)	3 (7.9%)	0.777
Anxiety disorder	6 (6.7%)	2 (3.8%)	4 (10.5%)	0.236
Conduct disorder	15 (16.7%)	13 (25%)	2 (5.3%)	**0.002**
ADHD	23 (25.6%)	12 (23.1%)	11 (28.9%)	0.528
**Personal psychiatric history before index suicide attempt**				
Previous psychiatric condition	45 (50%)	31 (59.6%)	14 (36.8%)	**0.03**
Personal history: affective disorders	20 (22.2%)	15 (28.8%)	5 (13.20%)	0.07
Personal history: anxiety disorders	11 (12.2%)	3 (5.8%)	8 (21.1%)	**0.03**
Personal history: conduct disorders	20 (22.2%)	17 (32.7%)	3 (7.9%)	**0.005**
Personal history: ADHD	14 (15.6%)	9 (17.3%)	5 (13.2%)	0.592
Prior suicide attempts	22 (24.4%)	16 (30.8%)	6 (15.8%)	0.1
Prior non-suicidal self-injuries	41 (45.6%)	23 (44.2%)	18 (47.4%)	0.768
**Family history of suicidal behavior**	15 (16.6%)	9 (17.3%)	6 (15.7%)	0.84
**Treatment with psychotropic drugs prior to the suicide attempt**				
Previous treatment with antidepressants	28 (31.1%)	17 (32.7%)	11 (32.7%)	0.705
Previous treatment with anxiolytics	19 (21.1%)	11 (21.2%)	8 (21.1%)	0.991
Previous treatment with psychostimulants	9 (10.0%)	6 (11.5%)	3 (7.9%)	0.569
Previous treatment with antipsychotics	7 (7.8%)	4 (7.7%)	3 (7.9%)	0.972
Scores of SIS (mean ± SD)	8.2 ± 5.6	8.3 ± 5.2	8.1 ± 6.2	0.904

*p*-values in bold are statistically significant (*p* < 0.05).

**Table 2 ijerph-20-02952-t002:** Generalized linear model (GLM) summary statistics performed on Beck’s Suicidal Intent Scale (SIS). Significant effect of continuous or categorical predictors are reported in bold.

	Wald Chi-Square	df	*p*-Value
(Intercept)	0.017	1	0.895
Current diagnosis (3 categories)	10.992	2	**0.004**
History of anxiety disorders	0.630	1	0.427
History of conduct disorders	0.416	1	0.519
Comorbid conduct disorders	0.100	1	0.751
Study period	0.233	1	0.629
Age	0.988	1	0.320
Current diagnosis × study period	0.878	2	0.645

## Data Availability

The data that support the findings of this study are available from corresponding authors upon reasonable request.

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
