# Peer review of "COVID-19 Pandemic Has Changed the Psychiatric Profile of Adolescents Attempting Suicide: A Cross-Sectional Comparison"

_ijerph, 2023, doi:10.3390/ijerph20042952_

Round 1

Reviewer 1 Report

"for committing SA" - better to write attempted suicide.. This is a contradiction of terms. How can a person commit and attempt suicide at the same time?

Died by suicide is now used instead of committed suicide. See https://www.thesaurus.com/e/ways-to-say/right-words-about-suicide/

It might be better to not use the acronym SA, since acronyms often create confusion.

While no plagiarism was found in the section I had checked, Grammarly found 16 writing issues - five grammar, four spelling, two conciseness, one word choice, and four others.

In the conclusion, you write "AS is the first detectable manifestation of the underlying disorder," Do you mean SA?

Comment on areas that your statistical analysis pointed out as significant. Why were they significantly different?

Reviewer 2 Report

The authors study a very important topic among adolescents, the clinical characteristics of adolescents attempting to suicide related to the COVID-19 pandemic.

There are some mistakes during statistical analysis which have a significant effect on the results and conclusions.

It is a cross-sectional study, but it was indirectly indicated only in the limitation section, please indicate this in the abstract and method section. Based on the study design, the conclusions are overstated, assuming changes based on the differences between pre-pandemic and pandemic groups. Please also correct the Title considering this fact.

In Abstract, Line 32 “During the pandemic, adolescents had no prior psychiatric history”. This contradicts the results and clinical variables in Table 1. In the Abstract, based on the GLM results, SA severity is associated with the current diagnosis, the question is whether this association was independent of the study period as mentioned in the conclusions. The aim of the study was, based on the Abstract, to assess the demographics and clinical characteristics of adolescents. The authors provide a lot of information about clinical characteristics, but demographic characteristics covered only gender and age. I think it is not an assessment of demographics.

Ethical considerations: was parental consent required or only yp consent?

Data analyses: Kolmogorov-Smirnov (please correct the name of the test) is an old-fashioned test for normality checking, and significance tests of normality are very sensitive to the sample size. I recommend that you use numeric methods, and report skewness and kurtosis to indicate if a variable is not normally distributed. In data analysis is mentioned the Mann-Whitney test but in the results, they also use the Kruskal-Wallis test. Did the age variable not normally distributed? Line 164-165, related to the GLM is not correct, there are limitations to the possible distributions, i.e; Poisson or binomial family. May SIS have Poisson distribution? The binomial distribution is correct for the clinical characteristics. Please correct p < 0.05 in Line 167. The level of significance was set, i.e., alpha, not the p-value which has not a threshold, it is a probability.

Results: Age differences between groups, only median was reported, based on this, I do not think there is an age difference between groups in practice. However, it would be an important issue if you detect a practically significant age difference. Please check mean and standard deviation (otherwise, parametric tests are robust to violation of normality, but not to inequality of variances).

In Lines 176-178, primary diagnosis at the time of assessment. Please correct the p-value for the primary diagnosis. One Fisher’s exact test relates to each psychiatric diagnosis (in crosstabs), but I do not think that the assumptions of the exact test met in each diagnosis because of the small cell sizes. Please check and correct these results. Add frequency data to the text to help the reader understand the results.

Do not use “did not reach statistical significance”, it is wordy and implies misunderstandings of the hypothesis testing; in statistics, a result can be statistically significant or non-significant.

Please add detailed results related to GLM, in a Table for all used variables. It is hard to understand the results, for example, what the rest of the factors referred to.  Line 210 and 211, I do not see connections between the sentences: no significant main effect and no significant interaction…indeed, there were significant differences? Figure 1 is completely wrong. First, it is not a histogram, it is a bar chart. You cannot visualize interaction or the lack of interaction with a bar chart. Furthermore, it is not a bar chart for SIS scores by diagnosis and period, you visualize only the means in two groups (period) by diagnosis. Please correct this figure and indicate SEM for Figure A too. A question was raised based on this result, what was the distribution of the SIS? You reported the result of a non-parametric test which is not based on the means, but you graphed the means of the groups in Figure 1. In this case, a frequency distribution is commonly used to show what the significant differences between groups mean.

In Lines 202-204, it is not correct. Were there any other reasons to recategorize the current diagnosis into three categories than the sample size? Please use rather a professional categorization or choose a more appropriate statistical procedure for your data. I do not think that it would be meaningful to put into the same category autism spectrum disorder and conduct disorder.

Based on my comments, I recommended that the authors should improve the results section and revise the Discussion section based on the results. Considering the study design, it is not recommended to overstate the conclusion.

Round 2

Reviewer 2 Report

I think all the necessary corrections have been made, I accept them. Thank you for your detailed replies and for the careful correction of the manuscript.